# You Only Live Once:
# Single-Life Reinforcement Learning

**Annie S. Chen**[1], **Archit Sharma**[1], **Sergey Levine**[2], **Chelsea Finn**[1]
Stanford University[1], UC Berkeley[2]
`asc8@stanford.edu`

## Abstract

Reinforcement learning algorithms are typically designed to learn a performant policy that can repeatedly and autonomously complete a task, usually starting from scratch. However, in many real-world situations, the goal might not be to learn a policy that can do the task repeatedly, but simply to perform a new task successfully once in a single trial. For example, imagine a disaster relief robot tasked with retrieving an item from a fallen building, where it cannot get direct supervision from humans. It must retrieve this object within one test-time trial, and must do so while tackling unknown obstacles, though it may leverage knowledge it has of the building before the disaster. We formalize this problem setting, which we call *single-life reinforcement learning* (SLRL), where an agent must complete a task within a single episode without interventions, utilizing its prior experience while contending with some form of novelty. SLRL provides a natural setting to study the challenge of autonomously adapting to unfamiliar situations, and we find that algorithms designed for standard episodic reinforcement learning often struggle to recover from out-of-distribution states in this setting. Motivated by this observation, we propose an algorithm, *Q*-weighted adversarial learning (QWALE), which employs a distribution matching strategy that leverages the agent's prior experience as guidance in novel situations. Our experiments on several single-life continuous control problems indicate that methods based on our distribution matching formulation are 20-60% more successful because they can more quickly recover from novel states. [1]

## 1 Introduction

Agents operating in the real world must often contend with novel situations that differ from their prior experience. For example, a search-and-rescue disaster relief robot may encounter novel obstacles while traversing a building. The vast range of situations that could be encountered at test time cannot be anticipated in advance during training, and therefore agents that can adapt at test time will be better equipped to cope with new situations. Reinforcement learning (RL) algorithms in principle provide a suitable paradigm for such adaptation. However, much of the work in RL focuses on learning an optimal policy that can repeatedly solve a given task, whereas on-the-fly adaptation simply requires the agent to complete the task once without human interventions or supervision, presenting a different set of challenges. In this work, we formalize this problem setting as *single-life reinforcement learning* (SLRL), where the agent is evaluated on its ability to successfully and autonomously adapt to a novel scenario within a single trial.

Given some prior experience, SLRL induces the challenge of contending with novelty (e.g., a new initial state distribution or new dynamics) at test time without episodic resets. A key problem that makes this difficult is that the agent will need to recover from unfamiliar situations without any

---

[1]Project website: `https://sites.google.com/stanford.edu/single-life-rl`

36th Conference on Neural Information Processing Systems (NeurIPS 2022).

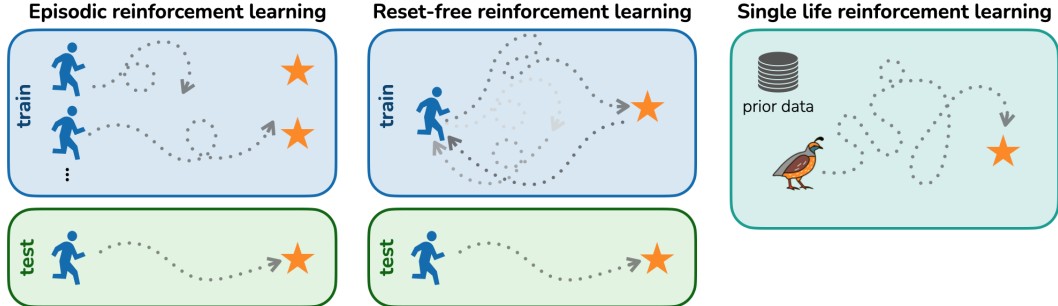

Figure 1: We study the single-life reinforcement learning (SLRL) problem, where given prior data, an agent must complete a task autonomously in a single trial in a domain with a novel distribution shift.

interventions, which can be especially challenging when informative shaped rewards are unavailable, as is often the case in the real world. In fact, task rewards are sometimes only provided at the end of the life, e.g. the search-and-rescue robot may only receive reward signal after it maneuvers to find a desired object and may therefore get stuck behind unknown debris. In episodic RL, the agent can rely on a reset to recover from an unfamiliar state. In contrast, in SLRL, the agent only has its prior experience to leverage in order to find its way back to a good state distribution. Consequently, we find that fine-tuning a pre-trained value function via online RL will struggle in SLRL settings, as it does not explicitly encourage the agent to return to its known distribution and complete the desired task.

To better handle the SLRL setting, we need a way to provide intermediate shaped rewards that guide the adaptation process, and help the agent recover when it gets stuck. One potential option for providing the desired guidance is to use reward shaping towards the agent's distribution of prior experience. Adversarial imitation learning (AIL) approaches such as GAIL (Ho and Ermon [2016]) can potentially provide such reward shaping, but using existing AIL methods naively may not give the intended behavior in the SLRL setting due to two main shortcomings. First, such methods assume expert demonstrations are given as prior data, but in SLRL, we may be given suboptimal offline prior data. Second, AIL methods train the agent to match the entire distribution of prior data, which may be key to learning an optimal policy, but may be a drawback in our setting, as the agent might not be consistently guided towards task completion. To address these shortcomings, we propose $Q$-weighted adversarial learning (QWALE), in which different states in the prior data are weighted different amounts by their estimated $Q$-value. QWALE incentivizes the agent to move towards states in the prior data with higher values, so that agent is more consistently guided towards states closer to task completion.

Our contributions are as follows. We formalize the SLRL problem setting, which provides a natural setting in which to study autonomous adaptation to novel situations in reinforcement learning. Moreover, we believe SLRL is a useful framework for modeling many situations in the real world. We explore reward shaping through distribution matching as one potential choice of methods for SLRL and identify challenges that uniquely arise in the SLRL setting with existing distribution matching approaches. We propose a new approach, $Q$-weighted adversarial learning (QWALE), which is less sensitive to the quality of prior data available and provides the agent with a shaped reward towards completing the desired task a single time. Through our experiments, we find that QWALE can meaningfully guide the agent to recover to the prior data distribution to complete the desired task 20-60% more successfully on four separate domains compared to existing distribution matching approaches and RL fine-tuning. QWALE provides a strong baseline for developing algorithms that can better adapt to novelty online and recover from out-of-distribution states.

## 2   Related Work

**Autonomous RL**. In the context of deep RL, agents typically (but not always) are trained in episodic setting and are evaluated on the quality of the learned policy. Several recent works have developed algorithms that can learn without episodic resets [Han et al., 2015, Eysenbach et al., 2017, Zhu et al., 2020a, Sharma et al., 2021a, 2022, Gupta et al., 2021, 2022, Kim et al., 2022, Sharma et al., 2021b]. Like our work, such methods aim to make it possible to learn without any episodic resets, but are typically still focused on acquiring an effective policy that can perform the task repeatedly, typically by training some auxiliary controller to enable the policy to "retry" the task multiple times without

resets. In contrast, our aim is to develop an algorithm that can solve the task once, but as quickly as possible, which introduces a unique set of challenges as we discussed above.

**Continual RL**. There is a rich literature on reinforcement learning in the continuing setting [Mahadevan, 1996, Schwartz, 1993, Sutton and Barto, 2018, Xu and Zhu, 2018, Rolnick et al., 2019, Lomonaco et al., 2020, Wei et al., 2020] that considers maximizing the average reward accumulated over an infinite horizon without episodic resets. Such works often also consider regret minimization as the objective. SLRL is closely related, and can be viewed as a special case where the agent has access to a prior offline dataset and aims to solve a single task as quickly as possible in a new domain with a distribution shift. While the focus in continual learning is on general "lifelong" methods or on exploration, our focus is on effectively leveraging prior data in a setting that is meant to be reflective of real-world tasks (for example, in robotics).

**Leveraging offline data in online RL**. Learning expert policies given prior interaction data has been extensively studied in imitation learning [Argall et al., 2009, Ross et al., 2011, Ghasemipour et al., 2020], inverse RL [Ng et al., 2000, Finn et al., 2016, Ziebart et al., 2008, 2010], RL for sparse reward settings [Brys et al., 2015, Nair et al., 2018, Rajeswaran et al., 2017, Hester et al., 2018, Vecerik et al., 2017] and offline RL [Wang et al., 2020b, Levine et al., 2020, Kumar et al., 2020, Kidambi et al., 2020, Wu et al., 2019a, Nair et al., 2018]. Across all these diverse topics, the goal is to learn a competent policy that can solve the task efficiently whereas the objective in this work is to complete the task in a single trial as quickly as possible. To this end, we build on recent adversarial approaches to inverse RL [Ho and Ermon, 2016, Fu et al., 2017, Singh et al., 2019, Torabi et al., 2019, Kostrikov et al., 2018, Zhu et al., 2020b] to encourage agent's state visitation towards expert prior data where the agent is likely to be successful. Prior methods have also studied adversarial inverse RL and imitation learning with non-expert data [Wang et al., 2018, Sun and Ma, 2019, Wu et al., 2019b, Wang et al., 2020a, 2021b,a, Cao et al., 2022, Beliaev et al., 2022]. However, as we discuss in Section 5, these approaches need to be adapted for the SLRL setting to be efficient at completing the task and to handle novelty, for example when the dynamics may have changed.

**Transfer and adaptation in RL**. Many prior works have studied the problem of adapting in presence of shifts between train and test settings, often in a specific problem setting such as sim2real transfer [Sadeghi and Levine, 2016, Tobin et al., 2017, Peng et al., 2018, Mehta et al., 2020] or fast adaptation via meta-learning Finn et al. [2017], Nichol et al. [2018], Nagabandi et al. [2018], Zintgraf et al. [2019], Finn et al. [2019]. A common theme in these works is that the algorithm can often train in preparation for adaptation at test-time, thus affecting the prior experiences it may collect. In contrast, the SLRL setting lays algorithmic emphasis on online exploration and adaptation, as the agent has access to a fixed prior dataset of experiences. Other transfer learning approaches adapt the weights of the policy to a new environment or task, either through rapid zero-shot adaptation Hansen et al. [2020], Yoneda et al. [2021] or through extended episodic online training Khetarpal et al. [2020], Rusu et al. [2016], Eysenbach et al. [2020], Xie et al. [2020], Xie and Finn [2021]. Unlike the latter, we focus on adaptation within a single episode, but, unlike the former, with a focus on extended exploration and learning over tens of thousands of timesteps. This problem setting leads to unique challenges, namely that the agent must autonomously recover from mistakes, hence requiring a distinct approach.

## 3  Preliminaries

In this section, we describe some preliminaries before formalizing our problem statement in the following section. We consider an agent that operates in a Markov decision process (MDP) consisting of the tuple $\mathcal{M} = (\mathcal{S}, \mathcal{A}, \mathcal{P}, \mathcal{R}, \rho_0, \gamma)$, where $\mathcal{S}$ is the state space, $\mathcal{A}$ is the agent's action space, $\mathcal{P}(s_{t+1}|s_t, a_t)$ represents the environment's transition dynamics, $\mathcal{R} : \mathcal{S} \to \mathbb{R}$ indicates the reward function, $\rho_0 : \mathcal{S} \to \mathbb{R}$ denotes the initial state distribution, and $\gamma \in [0, 1)$ denotes the discount factor. In typical reinforcement learning, the objective is find a policy $\pi$ that maximizes $J(\pi) = \mathbb{E}_{\tau \sim \pi}[\sum_{t=0}^{\infty} \gamma^t \mathcal{R}(s_t)]$.

Although our method is a reward-driven RL algorithm, we utilize concepts from imitation learning to overcome sparse rewards, utilizing potentially suboptimal prior data. To this end, we build on adversarial imitation learning (AIL), which uses prior data $\mathcal{D}_{\text{prior}}$ in the form of expert demonstrations (we will relax this requirement) to recover the expert's policy.

One such method is GAIL [Ho and Ermon, 2016], which finds a policy $\pi_\theta$ that minimizes the Jensen-Shannon divergence between its stationary distribution and the expert data. It does so by

training a discriminator network $D : \mathcal{S} \times \mathcal{A} \to (0, 1)$, alternating updates with updates to the policy $\pi$. Concretely, $D$ and $\pi$ are learned by optimizing the following:

$$\min_{\pi} \max_{D \in (0,1)^{\mathcal{S} \times \mathcal{A}}} \mathbb{E}_{\pi_E}[\log(D(s, a))] + \mathbb{E}_{\pi}[\log(1 - D(s, a))] - \lambda \mathcal{H}(\pi)$$
$$= \min_{\pi} \mathcal{D}_{\text{JS}}(\rho^{\pi}(s, a) \mid\mid \rho^*(s, a)) - \lambda \mathcal{H}(\pi), \tag{1}$$

where $\mathcal{D}_{\text{JS}}$ is the Jensen-Shannon divergence and $\rho^*(s, a)$ is the optimal state-action occupancy measure of the expert $\pi_E$ (represented by demonstrations). In Section 6, we will see that AIL-style discriminator-based approaches can be adapted to the SLRL problem setting *without* demonstration data, and will therefore form the basis of our method.

## 4  Single-Life Reinforcement Learning

In this section, we formalize the *single-life reinforcement learning (SLRL)* problem. The defining characteristic of SLRL is that the agent is given a single "life" (i.e., one long trial) to complete a desired task, with the trial ending when the task is completed. The agent must complete the task autonomously, without access to any human interventions or resets. Furthermore, as is the case in many real-world situations, when faced with situations where a task must be completed once, informative shaped rewards are not easily obtained, and so the single life might have a reward only once after the task has been completed. However, an agent typically has some prior knowledge. E.g., an agent tasked with retrieving a valuable from a burning building may have experience finding items in that building before the fire. We will therefore assume access to offline prior data of some sort that the agent may use for pretraining and during its single life. In many cases such as the disaster relief example, we may not have expert prior data of the desired task in the desired environment. Hence, we emulate this setting by providing the agent with prior data from a related environment and deploying its single life on a domain with a distribution shift.

We can formalize this setting as follows. We are given prior data $\mathcal{D}_{\text{prior}}$, which consists of transitions from some source MDP $\mathcal{M}_{\text{source}}$. The agent will then interact with a target MDP defined by $\mathcal{M}_{\text{target}} = (\mathcal{S}, \mathcal{A}, \tilde{\mathcal{P}}, \mathcal{R}, \tilde{\rho}_0, \gamma)$. We assume that the the target MDP has an aspect of novelty not present in the source MDP, such as different dynamics $\tilde{\mathcal{P}}(s_{t+1} \mid s_t, a_t)$ or a different initial state distribution $\tilde{\rho}$. Naturally, the more similar the domains are, the easier the problem becomes, and the effectiveness of any algorithm will be strongly dependent on the degree of similarity, though formalizing a precise assumption on similarity between the source and target domain is difficult. The reward between the two MDPs is the same, meaning the agent is still trying to accomplish the same task in the target domain as in the source.

Maintaining the same notation as the previous section, the SLRL problem aims to maximize $J = \sum_{t=0}^{h} \gamma^t \mathcal{R}(s_t)$, where $h$ is the trial horizon, which may be $\infty$. In general, we expect the task reward to be such that learning *only* from task rewards during the single life is difficult, as the reward may be very sparse or even awarded only upon successful completion of the task. We assume that there are no sink states beyond a terminal success state, such that it is possible for the agent to autonomously recover from any mistake. Note that this setup is essentially the same as the widely studied regret minimization problem in exploration [Mahadevan, 1996]. However, while regret minimization is typically studied in the context of RL exploration theory, our aim with SLRL is to study a particular *special case* of the more general regret minimization framework that is meant to reflect a realistic setting in real-world RL (e.g., robotics) where an agent with prior experience must solve a task in a single (potentially long) trial.

As we analyze in Section 6, algorithms designed for episodic policy learning do not perform well in the SLRL problem setting, even when the policy and replay buffer are pre-trained and seeded with the prior data, because they do not quickly recover from mistakes to get back onto a good state distribution. In Section 5, we will discuss an approach based on distribution matching that attempts to address this issue.

## 5  $Q$-weighted Adversarial Learning (QWALE)

In this section, we will present our method for addressing SLRL, which we call QWALE. The key insight in QWALE is to utilize the prior data $\mathcal{D}_{\text{prior}}$ to efficiently complete the task, especially when the reward information is sparse in the target domain. The framework of AIL provides a reasonable starting point, though it is not sufficient by itself: existing AIL methods assume that the prior data

consists of expert demonstrations, and they train the agent to match the entire demo distribution. Since our goal is not to learn a policy that repeatedly performs the task, but rather to solve it as quickly as possible once, we do not actually want to *learn to imitate* prior data, but rather to seek out states that resemble the *best* states in the prior data, with better states being more preferred. This is especially important when the prior data is not actually optimal, but might consist of arbitrarily suboptimal states. We will discuss how this can be accomplished with a modification of AIL which, instead of treating prior data as equally desirable, preferentially drives the agent toward states that resemble the best states in the prior data.

## 5.1 Algorithm Description

Our algorithm's desired behavior is to lead the agent towards the distribution of prior data, therefore helping it recover when it falls into out-of-distribution states, while also pushing it towards task completion. GAIL [Ho and Ermon, 2016] offers a framework to shape the state-action distribution towards that of an expert. In particular, GAIL minimizes $\mathcal{D}_{\mathrm{JS}}(\rho^\pi(s, a) \ || \ \rho^*(s, a))$ (ignoring the causal entropy $\mathcal{H}(\pi)$). In SLRL, we relax this assumption, as we may only have access to suboptimal offline prior data. As such, we desire an algorithm that will be agnostic to the quality of prior data. Hence, instead of learning a policy that aims to uniformly match the entire state-action distribution of the prior data, we want to match the state-action pairs that will lead to task completion.

The problem of matching the state-action distribution has been studied in stochastic optimal control, particularly in REPS [Peters et al., 2010]. Roughly, this problem can be framed as the RL problem with an additional constraint encouraging the learned distribution to be close to the distribution of the prior data. The solution to this optimization problem gives us that our desired target state-action distribution has the form $\rho^*_{\mathrm{target}}(s, a) \propto \rho_\beta(s, a) \exp(Q^{\pi_{\mathrm{target}}}(s, a) - V^{\pi_{\mathrm{target}}}(s))$, where the corresponding policy can be derived by using $\pi_{\mathrm{target}} = \rho^*_{\mathrm{target}}(s, a) / \int_{\mathcal{A}} \rho^*_{\mathrm{target}}(s, a') \mathrm{d}a'$ using [Syed et al., 2008]. For a formal description of the problem, see Appendix A.1.

Thus, replacing $\rho^*_{\mathrm{target}}$ as the target distribution instead of $\rho^*$, we propose $Q$-weighted adversarial learning (QWALE), which minimizes $\mathcal{D}_{\mathrm{JS}}(\rho^\pi \ || \ \rho^*_{\mathrm{target}})$ as follows:

$$\min_\pi \mathcal{D}_{\mathrm{JS}}(\rho^\pi(s, a) \ || \ \rho^*_{\mathrm{target}}(s, a)) = \min_\pi \max_D \mathbb{E}_{s, a \sim \rho^*_{\mathrm{target}}} \left[\log D(s, a)\right] + \mathbb{E}_{s, a \sim \rho^\pi} \left[\log(1 - D(s, a))\right]$$

$$= \min_\pi \max_D \mathbb{E}_{s, a \sim \rho^\beta} \left[ \frac{\exp(Q^{\pi_{\mathrm{target}}}(s, a) - V^{\pi_{\mathrm{target}}}(s))}{\mathbb{E}_{\rho^\beta} \left[\exp(Q^{\pi_{\mathrm{target}}}(s, a) - V^{\pi_{\mathrm{target}}}(s))\right]} \log D(s, a) \right] + \mathbb{E}_{s, a \sim \rho^\pi} \left[\log(1 - D(s, a))\right]$$

$$\equiv \min_\pi \max_C \mathbb{E}_{s, a \sim \rho^\beta} \left[\exp\left(Q^{\pi_{\mathrm{target}}}(s, a) - V^{\pi_{\mathrm{target}}}(s)\right) \log D(s, a)\right] + \mathbb{E}_{s, a \sim \rho^\pi} \left[\log(1 - D(s, a))\right],$$

where the constant $\mathbb{E}_{\rho^\beta} \left[\exp(Q^{\pi_{\mathrm{target}}}(s, a) - V^{\pi_{\mathrm{target}}}(s))\right]$ can be ignored.

Hence, QWALE trains a weighted discriminator using a fixed $Q$-function $Q(s, a)$ trained in the source MDP to distinguish between useful transitions and ones that may be less useful. This $Q$-function may be obtained through RL pretraining, which is what we use for our experiments, or a variety of other ways, such as offline RL or Monte Carlo estimation. We train the discriminator in a similar manner as GAIL, where the positives come from offline data and negatives from online experience.

When training the discriminator, we weight the positive states $s$ by $\exp(Q(s, a) - b)$, where $b$ is an implementation detail treated as a hyperparameter. Following Abdolmaleki et al. [2018], we exclude $V^{\pi_{\mathrm{target}}}(s)$ in the weighting term in our practical implementation. Including the term $b$ changes the bias on the discriminator, which in effect just adds a constant to the reward. In practice, to avoid having to tune $b$, we just use the value of the most recent state as $b$, i.e. $b = Q(s_t, a_t)$. We include additional discussion in the Appendix. We normalize the $Q$-values to be between 0 and 1 by subtracting the min and dividing by the (max - min) and train the $Q$-weighted discriminator in alternating updates that finetune the policy and critic in an AIL fashion. In this manner, QWALE extends AIL to the general setting with any prior data.

Intuitively, the goal of our weighted discriminator training procedure is to obtain a discriminator that, when used as a reward, will drive the agent toward states that it believes would lead to better outcomes than its present state, based on the prior data. The $Q$-function quantifies the agent's belief from the prior data that a particular state will lead to high reward, making this a natural choice for estimating how desirable a state is at any given time. Hence, using $Q$-values to weight the examples for the discriminator will cause the discriminator to prefer states that are closer to the goal over states that are further away. This is significantly different from the behavior we would expect to see if we

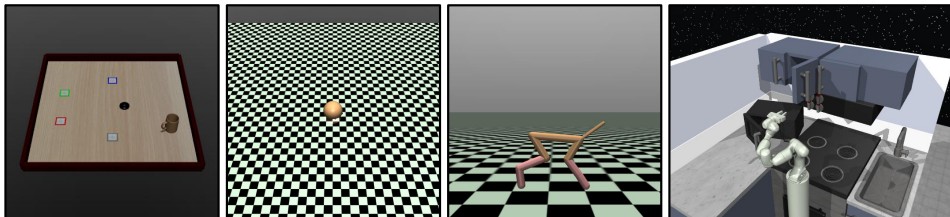

Figure 2: We evaluate in four different domains, including Tabletop-Organization, Pointmass, HalfCheetah, and a Franka-Kitchen environment with a microwave and cabinet. At test time, an aspect of novelty is introduced in each environment–new initial mug positions for Tabletop, wind for Pointmass, hurdles for the HalfCheetah, and a new combination of tasks for the Franka-Kitchen.

were simply imitating the optimal policy, which would give equal weight to all of the transitions along an optimal path.

## 5.2 Practical Implementation

---

**Algorithm 1** Q-WEIGHTED ADVERSARIAL LEARNING (QWALE)

---

1: // Single Trial Deployment
2: **Require:** $\mathcal{D}_{\text{prior}}$, test MDP $\mathcal{M}_{\text{test}}$, pretrained critic $Q(s, a)$, and (optionally) policy $\pi$;
3: **Initialize:** replay buffer for online transitions $\mathcal{D}_{\text{online}}$; parameters $\phi$ for discriminator $q_\phi(\text{prior} \mid s_t)$, timestep $t = 0$
4: **while** task not complete **do**
5:     Take $a \sim \pi(\cdot \mid s_t)$, and observe $r_t$ and $s_{t+1}$
6:     $\mathcal{D}_{\text{online}} \leftarrow \mathcal{D}_{\text{online}} \cup \{(s_t, a_t, r_t, s_{t+1})\}$
7:     $\phi \leftarrow \phi - \eta \nabla_\phi L(\phi)$ // Update discriminator according to Eq. 2
8:     $r'(s_t) = r(s_t) - \log(1 - q_\phi(\text{prior} \mid s_t))$
9:     $Q(s, a), \pi \leftarrow \text{SAC}(Q(s, a), \pi, \mathcal{D}_{\text{prior}} \cup \mathcal{D}_{\text{online}}, r')$
10:    Increment $t$

---

We optimize our objective using maximum entropy off-policy RL with the SAC algorithm (Haarnoja et al. [2018]), modified in a similar manner as we did with GAIL in the previous section. In particular, we learn an additional discriminator $q_\phi(\text{prior} \mid s_t)$, optimized using standard cross-entropy loss, which is weighted accordingly.

$$L(\phi) = -\mathbb{E}_{\mathcal{D}_{\text{prior}}}[\exp(Q(s, a) - b) \log q_\phi(\text{prior} \mid s)] - \mathbb{E}_{\mathcal{D}_{\text{online}}}[\log q_\phi(\text{online} \mid s)]. \quad (2)$$

The discriminator is used to modify the rewards when updating off-policy from all experience–prior and online. At single trial test time, the actor and critic are optionally initialized with the pretrained weights, and the replay buffer is initialized with the offline data. For details such as network architecture and hyperparameters, see Appendix A. We present the full algorithm in Algorithm 1.

## 6 Experiments

The goal of our experiments is to answer the following questions: (1) How does QWALE compare to prior reinforcement learning and distribution matching approaches in single-life RL settings? (2) Do distribution matching approaches help agents learn to recover from novel situations in single-life RL? (3) How does QWALE compare to different variants of adversarial imitation learning, with different prior datasets?

### 6.1 Experimental Setup

To answer the above questions, we construct four single-life RL domains with varying prior datasets and sources of novelty, and then measure performance both in terms of speed of task completion and overall single-life success. In this subsection, we describe this experimental set-up in detail.

**Environments**. We consider the following four problem domains. First, in the Tabletop-Organization environment from the EARL benchmark Sharma et al. [2021b], the agent is tasked with bringing a mug to one of four different locations designated by a goal coaster. The prior data always has the same starting position of the mug. In the target environment, the starting position is in a new location unseen in the prior data. Second, the Pointmass setting tasks an agent to move in 2D from

its starting location at the origin $(0, 0)$ to the point $(100, 0)$. The target environment introduces a dynamics shift in the form of a strong "wind", where the agent is involuntarily pushed upward in the y-direction each step. Third, we construct a modified HalfCheetah environment, in which it is difficult but feasible for the cheetah to recover when flipped over. The target environment includes hurdles that the cheetah must jump over, as the prior data does not include these obstacles. Finally, we evaluate on a modified Franka-Kitchen environment, adapted from Gupta et al. [2019], where the task is to close a microwave and a hinged cabinet. The prior data only contains trajectories of closing the microwave and the hinged cabinet separately, so the agent must figure out online how to complete both tasks in a row. In other words, both objects are open at the start of single-life RL, and the agent has only previously seen instances where only one is open. For the latter two environments, dense rewards are given, and the discriminator-based reward is added to the extrinsic reward during single-life training. These environments are shown in Figure 2. Further details on the environments are given in Appendix A.

**Comparisons.** To answer question (1), we compare QWALE to three alternative methods: (a) SAC fine-tuning, which pre-trains a policy and value function in the source setting and fine-tunes for a single, long episode in the target environment, (b) SAC-RND, which additionally includes an RND exploration bonus Burda et al. [2018] during single-life fine-tuning, and (c) GAIL-s, which runs generative adversarial imitation learning Ho and Ermon [2016], Kostrikov et al. [2018] where the discriminator only operates on the current state $s$. We choose for the discriminator to only look at $s$ so that it is less susceptible to dynamics shift between the source data and target environment. We additionally compare to GAIL-sa, which passes both the current state and action to the discriminator. All methods use soft actor-critic (SAC) Haarnoja et al. [2018] as the base RL algorithm. We present additional comparisons in the Appendix.

**Prior datasets**. For all four environments, we evaluate SLRL using data collected through RL as our prior data. More specifically, we run SAC in the source MDP in the standard episodic RL setting for $K$ steps and take the last 50,000 transitions as the prior data. $K$ is chosen such that the prior data contains some good transitions but has not converged to an optimal policy yet. While we are able to run episodic RL in the source MDP, this is not a requirement for SLRL, as long as prior data in the source MDP is available. For all methods, including QWALE, GAIL variants, and SAC fine-tuning variants, the policy and value function are pretrained in this manner for the initialization of single-life RL. We note that AIL methods like GAIL typically assume that the prior data consists of expert demonstrations but we apply the algorithm only using mixed quality prior data, unless otherwise noted. In particular, Section 6.4 further evaluates AIL methods using demos as prior data, using 10 demonstrations for the Tabletop environment and 3 demonstrations for the Pointmass domain. We include such experiments to answer question (3), i.e. to investigate how the quality of prior data may affect performance.

**Evaluation Metrics.** To evaluate each method in each environment, we report the average and median number of steps taken before task completion across 10 seeds along with the standard error and success rate (out of 10). During single-life RL, for all environments, the agent is given a maximum of 200,000 steps to complete the task. If it has not completed the task after 200k steps, then 200k is logged as the total number of steps, and the run is marked as unsuccessful.

## 6.2 Results using mixed data as prior data

In this subsection, we aim to answer our first experimental question and study how QWALE performs compared to prior reinforcement learning and distribution matching approaches in single-life RL settings. As seen in Table 1, we find that QWALE achieves the lowest average and median number of steps as well as highest number of successes on all four domains. In particular, QWALE takes 20-40% fewer steps on average as the next best performing method. GAIL-s and GAIL-sa also outperform finetuning SAC across three of the four domains. These results demonstrate the suitability of distribution-matching approaches over RL finetuning in the SLRL setting. We see that guidance particularly towards a good state distribution is important, as we compare to finetuning SAC with an exploration bonus through random network distillation (Burda et al. [2018]). From Table 1, although RND may improve performance, particularly in the Tabletop domain, it generally does not perform as well as the distribution matching approaches, especially QWALE, showing that simply increasing exploration is not enough. Furthermore, these results show that the additional shaping provided by weighting the prior data by $Q$-value when training the discriminator can significantly improve guidance towards the goal. While GAIL gives equal weight to all transitions along an optimal path, the agent in QWALE is consistently guided towards states in the prior data with higher

| | Method | Avg ± Std error | Success / 10 | Median | | Method | Avg ± Std error | Success / 10 | Median |
|---|---|---|---|---|---|---|---|---|---|
| Tabletop | GAIL-s | 83.2k ± 23.8k | 8 | 75.6k | Cheetah | GAIL-s | 99.2k ± 23.0k | 7 | 77.4k |
| | GAIL-sa | 61.5k ± 28.7k | 7 | **2.4k** | | GAIL-sa | 102.0k ± 19.3k | 8 | 85.6k |
| | SAC | 123.7k ± 25.5k | 7 | 157.2k | | SAC | 128.9k ± 23.9k | 5 | 138.2k |
| | SAC-RND | 94.8k ± 26.9k | 7 | 51.4k | | SAC-RND | 158.4k ± 20.5k | 3 | 200.0k |
| | QWALE (ours) | **44.4k ± 24.6k** | **9** | 8.9k | | QWALE (ours) | **74.3k ± 9.5k** | **10** | **68.9k** |
| Pointmass | GAIL-s | 100.9k ± 33.0k | 5 | 101.9k | Kitchen | GAIL-s | 111.3k ± 27.9k | 6 | 122.1k |
| | GAIL-sa | 140.4k ± 30.3k | 3 | 200.0k | | GAIL-sa | 127.8k ± 26.9k | 5 | 189.1k |
| | SAC | 198.1k ± 1.9k | 1 | 200.0k | | SAC | 112.5k ± 25.8k | 6 | 122.3k |
| | SAC-RND | 184.1k ± 15.9k | 1 | 200.0k | | SAC-RND | 132.7k ± 26.2k | 4 | 200.0k |
| | QWALE (ours) | **61.0k ± 28.8k** | **8** | **1.7k** | | QWALE (ours) | **79.9k ± 21.7k** | **8** | **59.3k** |

Table 1: We evaluate the performance of QWALE to finetuning SAC and GAIL in our four environments using mixed data collected through RL as prior data. We omit the results of Behavior Cloning (BC), as it is unsuccessful at completing the task in every domain due to the distribution shift. We find that GAIL outperforms finetuning SAC and SAC-RND in 3 out of 4 domains, and QWALE outperforms both GAIL variants on all 4 domains on the average number of steps before task completion. All methods are evaluated over 10 seeds.

$Q$-value, leading to more efficient and reliable single-life task completion. Additional comparisons are presented in the Appendix.

### 6.3 Recovering from novel situations in SLRL

Next, to answer question (2), we analyze how QWALE helps agents learn to recover from novel situations in SLRL. First, we analyze the state visitation of SAC finetuning in the online phase for the Tabletop and Pointmass domains. A key challenge of SLRL (and also fully autonomous RL in general) is that if the agent falls off of the distribution, it cannot rely on resets to get back on track. Since there is a gap between the source and target domains, the agent will inevitably find itself in states that are out of distribution from the prior data. A value function pre-trained on the source data could in principle be used to evaluate states and guide the agent back towards good states, but it will be inaccurate on states outside of the prior data Fu et al. [2019]; then, when the value of some of those out-of-distributions states is overestimated, the value function may misguide the policy away from good states. Hence, fine-tuning a pre-trained value func-

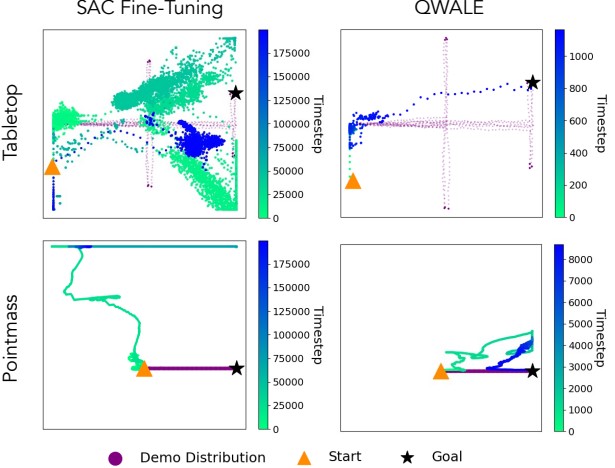

Figure 3: We visualize the online state visitation of the mug in the Tabletop (top) and the agent in the Pointmass (bottom) environments of a single-life trial using SAC finetuning (left) and QWALE (right). The expert states are in purple, and we color the trajectories green to blue according to timestep for both methods. See Section 5 for plots colored according to reward. In both environments, the SAC fine-tuning agent fails to recover from novel states, but is guided towards task completion with QWALE.

tion via online RL will not explicitly encourage the agent to get back on distribution, especially in sparse reward envs. As a result, the agent may spend a lot of time (perhaps infinite time) drifting once it falls out of distribution, which we see occurs in Figure 3.

On the other hand, distribution-matching methods like QWALE will explicitly encourage the agent to get back on distribution, by giving higher rewards on distribution than off distribution. In Figure 3, we visualize QWALE's state visitation in the Tabletop and Pointmass domains throughout a single lifetime. We color the trajectories according to timestep for both methods, and include the trajectories colored by reward (discriminator score) in Figure 5. The coloring in the timestep-colored plots is highly correlated with that in the reward-colored plots, showing how the reward helps the agent recover from unfamiliar states and gradually guides the agent towards the goal. In particular, when the agent is out of distribution, the agent is incentivized to explore states that will lead it closer back

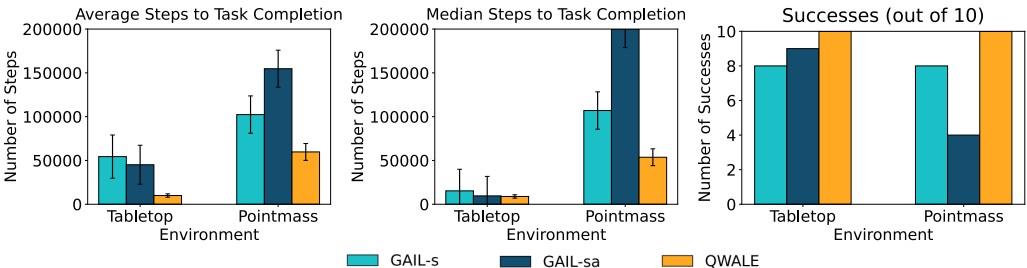

Figure 4: Discriminator-based approaches to single-life RL with expert demonstrations as the prior data. Given demonstration data, in both the Tabletop and Pointmass domains, QWALE matches or significantly outperforms both GAIL variants on all metrics.

to the prior state distribution, and when the agent is within distribution, it is incentivized to move to states closer to the goal, leading to efficient task completion.

### 6.4    Using demos as prior data

Finally, we evaluate the performance of different discriminator-based approaches in the two SLRL problem settings–Tabletop and Pointmass–where demonstration data is available as prior data. We compare using AIL with a state-only discriminator (GAIL-s) as well as with a state-action discriminator (GAIL-sa) to our proposed $Q$-weighted discriminator method (QWALE). With the latter method, we have access to the same $Q$-function pretrained when collecting prior data using standard RL for the mixed data experiments above, but we do not initialize any of the algorithms at test time with the pretrained policy and critic weights.

From Figure 4, the GAIL variants are both able to consistently solve the task in the Tabletop domain, but unsurprisingly, GAIL-sa does especially poorly in the Pointmass domain, where the dynamics have changed at test time. Compared to the two GAIL variants, QWALE gives a significant improvement in both domains. These results demonstrate how more detailed reward shaping towards the completion of the desired task can be helpful in the SLRL setting even with demonstrations as prior data. Moreover, comparing these results with those in Table 1, while access to expert data as prior data unsurprisingly improves the performance of GAIL methods, it can also improve the performance of QWALE.

## 7    Conclusion

In this paper, we formalized and studied the problem setting of single-life reinforcement learning: a setting where an agent needs to autonomously complete a task once in a single trial while drawing upon prior experience from a related environment. We found that standard fine-tuning via RL is ill-suited for this problem because the algorithm struggles to recover from mistakes and novel situations. We hypothesized that this happens since resets in episodic RL prevent algorithms from needing to recover, whereas single-life RL and continuing settings in general do demand the agent to find its way back to good states on its own. We then postulated that distribution matching methods that aim to match the distribution of related prior data may help agents recover via reward shaping, and presented a new distribution matching method, QWALE, that weights examples by their $Q$-value. Our experiments verified that distribution matching approaches indeed do make better use of prior data, and that QWALE outperforms prior distribution matching methods on four single-life RL problems.

While QWALE can efficiently complete novel target tasks in a single episode without any interventions, important limitations remain. No algorithm, including QWALE, was able to complete the target task with 100% success, indicating that future works should aim to improve an algorithm's ability to solve tasks consistently. Moreover, the methods that we evaluated all used a pre-trained policy and value function from the source domain, which may be difficult to obtain in some source scenarios, as opposed to only obtaining some demonstrations or offline data. Finally, it would be interesting to explore problems with greater degrees of novelty between the source and target environments. In our experimental domains, the reward and state space between the source and target MDPs were assumed to stay the same. Eventually, we hope that this work may be extended to the case where the reward may be different in the online trial from the prior data. We expect that such settings would place even greater importance on autonomy and exploration, requiring sophisticated strategies for adapting

online to novelty. We hope that future work can explore these interesting questions and continue to make progress on allowing RL agents to autonomously complete tasks within a single lifetime.

## Acknowledgments

We thank members of the IRIS lab for helpful discussions on this project. This research was supported by Google and the Office of Naval Research (ONR) via grants N00014-21-1-2685 and N00014-20-1-2675. Annie Chen is supported by an NSF graduate student fellowship. Chelsea Finn is a CIFAR fellow.

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
