# A Appendix

## A.1 Additional Method Justification

The key idea of QWALE is to lead the agent to nearby states within distribution of the prior data if it is out of distribution and to nearby states closer to task completion if in distribution. GAIL finds a policy whose occupancy measure minimizes the Jensen-Shannon divergence $\mathcal{D}_{\mathrm{JS}}$ to the prior data. Since our objective in SLRL is to finish the task as soon as possible, and we may not be given expert demonstrations as prior data, we want to match the state-action pairs to those that lead to task completion.

To this end, we consider the following optimization problem:

$$\max_{\rho} \mathbb{E}_{\rho}\left[r(s,a)\right] = \max_{\rho} \int_{\mathcal{S}\times\mathcal{A}} \rho(s,a)r(s,a)\mathrm{d}s\mathrm{d}a \tag{3}$$

$$\text{s.t.} \quad \int_{\mathcal{S}\times\mathcal{A}} \rho(s,a)\mathrm{d}s\mathrm{d}a = 1, \quad \rho(s,a) \geq 0 \quad \forall (s,a) \in \mathcal{S} \times \mathcal{A}, \tag{4}$$

$$\int_{\mathcal{A}} \rho(s,a)\mathrm{d}a = (1-\gamma)\rho_0(s) + \gamma \int_{\mathcal{S}\times\mathcal{A}} \rho(s',a')\mathcal{P}(s \mid s',a')\mathrm{d}s'\mathrm{d}a' \quad \forall s \in \mathcal{S}, \tag{5}$$

$$\mathrm{KL}\left(\rho(s,a) \,||\, \rho_{\beta}(s,a)\right) \leq \epsilon. \tag{6}$$

Equations ( 3- 5) represent the reinforcement learning problem stated in the formalism of convex optimization, where $\rho$ represents the state-action distribution being optimized, Eq 4 constrains $\rho$ to be a valid probability distribution and Eq 5 constrains $\rho$ to be consistent with the MDP. The state-distribution $\rho^*$ of the expert policy $\pi_E$ is a solution of this problem. By introducing the constraint in Eq 6, we want to encourage the optimal distribution $\rho^*_{\mathrm{target}}(s,a)$ to be *soft* by being close to the prior state-action distribution $\rho_{\beta}$ (the state-action distribution of $\mathcal{D}_{\mathrm{prior}}$ corresponding to some behavior policy $\beta$). This problem has been studied in stochastic optimal control, particularly REPS [Peters et al., 2010]. REPS shows that the solution to the optimization problem in Eq( 3- 6) has the form $\rho^*_{\mathrm{target}}(s,a) \propto \rho_{\beta}(s,a)\exp(Q^{\pi_{\mathrm{target}}}(s,a) - V^{\pi_{\mathrm{target}}}(s))$, where the corresponding policy can be derived by using $\pi_{\mathrm{target}} = \rho^*_{\mathrm{target}}(s,a)/\int_{\mathcal{A}} \rho^*_{\mathrm{target}}(s,a')\mathrm{d}a'$ from [Syed et al., 2008].

## A.2 Implementation Details and Hyperparameters

In our experiments, we use soft actor-critic [Haarnoja et al., 2018] as our base RL algorithm. We use default hyperparameter values: a learning rate of 3e-4 for all networks, optimized using Adam, with a batch size of 256 sampled from the entire replay buffer (both prior and online data), a discount factor of 0.99. The policy and critic networks are MLPs with 2 fully-connected hidden layers of size 256. For all methods training a discriminator, it is parameterized as an MLP with 1 fully-connected hidden layer of size 128 and trained with a batch size of 512. During the online trial, 1000 steps are taken as initial collection steps before network updates begin. For all methods training a discriminator, we use mixup regularization Zhang et al. [2017] to reduce the brittleness of the discriminator.

Following [Sharma et al., 2021b], we use a biased TD update, where $Q(s_t,a_t) \leftarrow r(s_t,a_t) + \gamma Q(s_{t+1},a_{t+1})$ if $t$ is not a multiple of 100, and $Q(s_t,a_t) \leftarrow r(s_t,a_t)$ if it is. We use this update for all our evaluated methods online in order to improve stability. Since the online trial of single-life RL may have a large training horizon with hundreds of thousands of steps, this may lead to unstable bootstrapping, as for each $t$, $Q(s_t,a_t)$ bootstraps on $Q(s_{t+1},a_{t+1})$. Following [Sharma et al., 2022], for the auxiliary reward given by the discriminator $D$, we use $r(s,a) = -\log(1 - D(s))$ instead of $r(s,a) = \log D(s)$ to further improve stability.

For QWALE, the weighting of states is offset by a value $b$. This value may be treated like a constant hyperparameter and tuned. Adding this value changes the bias on the discriminator, which in effect adds a constant to the reward, though that constant changes over the course of training. In practice, to avoid having to tune $b$, we just use the value of the most recent state as $b$, i.e. $b = Q(s_t,a_t)$. To better interpret this value, with this weighting, $b$ is a baseline value capturing some notion of current progress. Prior data tends to get small weights if they have worse value than the current state, so the agent is consistently incentivized to move towards states with higher value than its current state.

For all experiments using prior data collected through RL, the agent was initialized at test time with the pretrained policy and critic. For QWALE, a copy of that critic was frozen and used when

calculating the weights for discriminator training. For all of the experiments with demonstration data in Section 6.4, the policy and critic were not initialized with any pretrained weights.

## A.3    Environment & Evaluation Details

*Tabletop-Organization.* The details for this environment are in [Sharma et al., 2021b]. The state space consists of the gripper's $(x, y)$ position, the mug's $(x, y)$ position, the gripper's state (whether attached to the mug or not), and the current goal, for a total of 12 dimensions. The action space is 3 dimensional, consisting of a delta in the gripper's $(x, y)$ position as well as an automatic gripper that will attach to the mug if the gripper is close enough. The tabletop extends from -2.8 to 2.8 in both the $x$ and $y$ directions. In the prior data, which consists either of 10 demonstrations or 50000 transitions collected through RL after 350000 steps of training, the initial state always places the mug at position (2.5, 0.0), and the goal is to place the mug at one of the following locations: (-2.5, -1.0), (-2.5, 1.0), (0, 2.0), (0, -2). For the online trial when evaluating SLRL, the mug is placed either at (2.7, 1.5) or (2.7, -1.5) with additional uniform randomness between (-0.15, 0.15) in both directions. This environment is also goal-conditioned at test time and the goal is randomly set to be either (-2.5, -1.0) or (-2.5, 1.0). The reward is 1 when the mug is within 0.15 distance of its goal position (at which point the single life ends) and 0 everywhere else.

*Pointmass.* The Pointmass environment has a 6-dimensional state space consisting of the agent's $(x, y)$ position, its $(x, y)$ velocity, and the $(x, y)$ coordinates of the goal. The environment extends between -100 and 100 along the $x$ axis and between -200 and 200 along the $y$ axis. The action space is 2-D, consisting of the delta in both directions, clipped between -1 and 1 for a single action. The prior data consists of 3 demonstrations or 50000 transitions collected through RL after 350000 steps of training. The agent starts at (0, 0) and the goal is at (100, 0) for the prior data and online trial. During the online trial, a strong "wind" is introduced, where a random amount between 0.8 and 0.9 is added to the agent's $y$ coordinate and 0.2 is subtracted from the agent's $x$ coordinate at each step. The reward is 1 when the agent is within a distance of 2 of the goal position and 0 everywhere else.

*HalfCheetah.* The HalfCheetah environment has a state space with 18 dimensions, consisting of the position and velocity of each joint. The prior data consists of 50000 transitions collected through RL after 150000 steps of training. The reward is $r_t = \Delta x_t - 0.1 * ||a_t||_2^2$. At test time, 10 hurdles are included in the environment, spread between the x-coordinate of 7 and 260. The cheetah starts at 0 and its single life is considered successful when it gets to the coordinate 300, although the information about the hurdles or goal are not included in the state space.

*Franka-Kitchen.* The Franka-Kitchen is adapted from [Gupta et al., 2019, Sharma et al., 2021b]. The state space consists of a 9 DoF position-controlled Franka-robot with a microwave and hinged cabinet. The prior data consists of 50000 transitions collected through standard episodic RL after 950000 steps of training, where one of the microwave or cabinet is open, and the task is to close that object. At test time, both are open, and the task is to close both objects. The reward function is equal to the sum of the Euclidean distance between the objects and their goal positions and the distance between the arm and its goal position.

## A.4    Additional Experiments

We run three additional variants of SAC finetuning on all four experimental domains using mixed prior data. More specifically, we compare to (a) SAC-no online, in which we pretrain SAC in the source environment and evaluate in the target domain without additional finetuning, (b) SAC-scratch, where we train SAC completely from scratch in the target domain during its single episode, and (c) SAC-BC, where we fine-tune SAC like the comparison in Section 6 but add an additional behavior cloning loss on the offline data. QWALE performs significantly better than three additional variants across all four domains. We find that SAC-no online performs poorly on all four test domains, which is unsurprising due to the distribution shift between the source and target environments. Adding a behavioral cloning loss does help in some domains but only if the dynamics are not changed–the method struggles in the Pointmass environment due to the changing dynamics at test time. SAC-scratch struggles compared to QWALE and distribution-matching approaches but sometimes performs better than finetuning SAC, such as in the Tabletop domain, showing that the distribution shift is so significant in some domains that pretraining in the source domain is counterproductive when deployed in the target domain.

In addition, we run a comparison with a state-of-the-art reset-free algorithm, MEDAL Sharma et al. [2022], as well as an additional comparison with Progressive Networks Rusu et al. [2016], a strong continual learning method, in the SLRL setting in the Tabletop domain. We include these comparisons

|  | Method | Avg ± Std error | Success / 10 | Median |  | Method | Avg ± Std error | Success / 10 | Median |
|---|---|---|---|---|---|---|---|---|---|
| Tabletop | SAC-no online | 181.5k ± 17.9k | 2 | 200.0k | Cheetah | SAC-no online | 143.7k ± 28.6k | 3 | 200.0k |
|  | SAC-scratch | 99.5k ± 22.5k | 8 | 98.8k |  | SAC-scratch | 200.0k ± 0 | 0 | 200.0k |
|  | SAC-BC | 80.8k ± 32.5k | 6 | 1.5k |  | SAC-BC | 93.5k ± 23.6k | 7 | 60.6k |
| Pointmass | SAC-no online | 200.0k ± 0 | 0 | 200.0k | Kitchen | SAC-no online | 200.0k ± 0 | 0 | 200.0k |
|  | SAC-scratch | 120.8 ± 18.5k | 8 | 112.9k |  | SAC-scratch | 155.0k ± 24.5k | 3 | 200.0k |
|  | SAC-BC | 200.0k ± 0 | 0 | 200.0k |  | SAC-BC | 101.1k ± 28.1k | 6 | 71.7k |

Table 2: We evaluate the performance of three additional SAC variants on our four experimental domains using mixed data collected through RL as prior data. QWALE significantly outperforms all variants on all 4 domains on the average number of steps before task completion. We find that SAC without online learning performs particularly poorly and finetuning SAC with a behavioral cloning loss only helps if the dynamics are not changed. All methods are evaluated over 10 seeds.

|  | Method | Avg ± Std error | Success / 10 | Median |
|---|---|---|---|---|
| Tabletop | MEDAL | 176.2k ± 15.9k | 3 | 200.0k |
|  | Progressive Networks | 159.0k ± 21.2k | 3 | 200.0k |

Table 3: We evaluate the performance of a state-of-the-art reset-free algorithm, MEDAL Sharma et al. [2022], and a continual learning approach, Progressive Networks Rusu et al. [2016], in the SLRL setting in the Tabletop domain. We find that both methods are not geared towards completing the desired task as quickly as possible.

to demonstrate how the SLRL setting differs from typical autonomous RL and continual learning settings. Both methods perform poorly in SLRL setting, which is unsurprising because SLRL has the goal of completing the desired task a single time as quickly as possible, which is different from the other two settings. Thus, typical reset-free and continual learning algorithms may not be effective in the single-life setting.

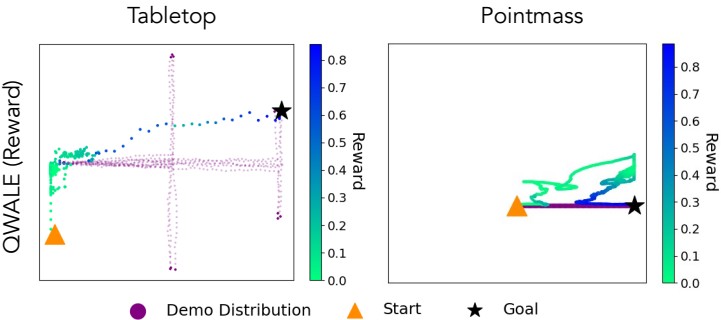

Figure 5: We visualize the online state visitation of the mug (w/ mirrored x-coordinate) in the Tabletop (left) and the agent in the Pointmass (right) environments of a single-life trial with QWALE. The expert demo states are colored in purple, and we color the trajectories green to blue according to reward. The weighted discriminator allots higher reward to states closer to the goal.

Finally, to supplement Figure 3, we visualize the online state visitation of the mug in the Tabletop and the agent in the Pointmass environments of a single-life trial with QWALE in Figure 5. These plots show how QWALE helps the agent recover from out-of-distribution states, as the reward gradually increases as the agent gets closer to the goal.