# OpenReview forum: "You Only Live Once: Single-Life Reinforcement Learning"
_NeurIPS.cc/2022/Conference — NeurIPS 2022 Accept_

### Official Review · Reviewer_bMAN · 2022-06-16

**Rating:** 6
**Confidence:** 3
**Soundness:** 3 good
**Presentation:** 3 good
**Contribution:** 3 good

**Summary:**

The paper introduces the setup of single-life reinforcement learning (SLRL), where the algorithm is given offline data from some environment and then is deployed in a related but slightly different environment; it needs to perform as good as possible in a single trial, leveraging online learning as well as offline data. This differs from both no-reset RL (because we care only about a single trial) and 0-shot generalization in RL (because we can learn along the way).

Then the authors propose QWALE, an algorithm based on GAIL, to tackle the setup of SLRL. They provide an empirical evaluation in 3 different continuous control environments. They compare their approach to several baselines, such as vanilla GAIL, RND, and vanilla SAC. They demonstrate that typically success rate is improved and, more visibly, the speed of reaching the solution is higher.


**Questions:**

[Q1] I have suggestions for additional baselines:
- just run SAC in the source environment and evaluate in the target one (without online learning); this is just a sanity check to verify that the shift is challenging enough - the performance should be very poor
- just train SAC on the target environment, completely from scratch
- fine-tune SAC, and add a behavioral cloning loss of the offline data as a regularization; this is another way to “anchor” the solution in the prior data. I conjecture this should help if only start states are changed (and not the dynamics), and otherwise would probably not work well.

[Q2] I’d suggest providing the details about the algorithm clearly in the main text, including the baseline term b described as “implementation detail” and details of normalization of Q values.

[Q3] I’d suggest citing the CRR work (mentioned in W2, section Weaknesses).


**Limitations:**

Please see W1 (section Weaknesses) and address this in the text.

**Strengths And Weaknesses:**

Strengths:

[S1] The introduced setup of SLRL seems important, describing a quite natural real-world scenario.

[S2] The paper is well written and easy to follow.

[S3] The empirical evaluation is sound and demonstrates improvements brought by the presented approach (see the Questions section for the discussion about baselines though).

Weaknesses:

[W1] The approach assumes that although there is a shift between source and target environments, the state space is essentially the same. If e.g. visual observations were used and the environment shifted from Earth to Mars, then the discriminator would learn to discriminate based on visual cues, which is not what we want in this approach. I think this is fine, it should be clearly stated as a limitation though.

[W2] The novelty of the method itself is limited. It is based on GAIL and adds a weighting based on the critic network, which is reminiscent of critic filtering in CRR ( https://arxiv.org/abs/2006.15134 ). However, it is used in a novel setup.

---

> ### Author Response · Authors · 2022-08-02
> **Response to Reviewer bMAN**
>
> We thank you for your detailed suggestions. We include clarifications and answers to individual points below. Please let us know if these responses address all of your concerns!
>
> > "The approach assumes that although there is a shift between source and target environments, the state space is essentially the same…"
>
> We have revised Section 7 to better clarify QWALE’s limitations with regards to assuming that the state space between the source and target MDPs stay the same. In our experimental setup, while the state space generally stays the same between source and target environments, their occupancies are not the same, e.g. it is inevitable that the agent will find itself in a state out of distribution from the prior data. For example, at test time in the Franka-Kitchen domain, both the microwave and cabinet are open, which was not encountered in the prior data.
>
> > "The novelty of the method itself is limited. It is based on GAIL and adds a weighting based on the critic network, which is reminiscent of critic filtering in CRR ( https://arxiv.org/abs/2006.15134 )"
>
> We have added a reference to CRR in the revised Section 2.
>
> > I have suggestions for additional baselines…
>
> Thank you for these experiment suggestions. We have run these additional baselines on all four domains and added their results to Table 2 in the Appendix and display them below. We find that the results are as you predicted, and we find that QWALE outperforms these additional baselines on all four domains.
>
> |                               | Method        | Avg $\pm$ Std error  | Success / 10 | Median |        | Method        | Avg $\pm$ Std error  | Success / 10 | Median |
> |-------------------------------|---------------|----------------------|--------------|--------|-----------------------------|---------------|----------------------|--------------|--------|
> | Tabletop  | SAC-no online | 181.5k   $\pm$ 17.9k | 2            | 200.0k | Cheetah | SAC-no online | 143.7k   $\pm$ 28.6k | 3            | 200.0k |
> |                               | SAC-scratch   | 99.5k  $\pm$ 22.5k   | 8            | 98.8k  |                             | SAC-scratch   | 200.0k $\pm$ 0       | 0            | 200.0k |
> |                               | SAC-BC        | 80.8k  $\pm$ 32.5k   | 6            | 1.5k   |                             | SAC-BC        | 93.5k $\pm$ 23.6k    | 7            | 60.6k  |
> | Pointmass | SAC-no online | 200.0k  $\pm$ 0      | 0            | 200.0k | Kitchen | SAC-no online | 200.0k  $\pm$ 0      | 0            | 200.0k |
> |                               | SAC-scratch   | 120.8  $\pm$ 18.5k   | 8            | 112.9k |                             | SAC-scratch   | 155.0k $\pm$ 24.5k   | 3            | 200.0k |
> |                               | SAC-BC        | 200.0k  $\pm$ 0      | 0            | 200.0k |                             | SAC-BC        | 101.1k $\pm$ 28.1k   | 6            | 71.7k  |
>
>
> > I'd suggest providing the details about the algorithm clearly in the main text…
>
> We have revised Section 5 to provide additional details for QWALE in the main text, including more information about the hyperparameter $b$ and normalization of Q values.

---

> > ### Comment · Reviewer_bMAN · 2022-08-08
> > **Response to the authors**
> >
> > Thank you for the very thorough revision of the work and responses to the reviewers. I think the paper would be an interesting addition to the RL literature. I feel like my original score still reflects my opinion about the paper, therefore I've decided to keep it.

---

> ### Author Response · Authors · 2022-08-05
> **Checking in**
>
> Hi reviewer bMAN,
>
> Thanks again for your thorough review! Please let us know if our response has addressed your concerns and if there's anything else you have questions about.

---

### Official Review · Reviewer_jw3X · 2022-07-09

**Rating:** 5
**Confidence:** 2
**Soundness:** 3 good
**Presentation:** 2 fair
**Contribution:** 2 fair

**Summary:**

The paper motivates single life RL setup involves pre-training on prior experience and generalizing another setting with novel states and that it is different from episodic or reset-free RL. As the objective is to perform well in single life on novel setting, the paper proposes to bias the exploration at test time to the known distribution at training time. For this, GAIL with the discriminator weighted with Q values while training on prior data is used. QWALE trained policy generalizes to new initial position of mug in Tabletop organization, wind in pointmass, hurdles in half cheetah and new combination of task in Franka-Kitchen environment.

**Questions:**

- SLRL is claimed to be a special case of continual or lifelong learning. SLRL focuses to explore as shown in the prior data and avoids exploring unknowns in the novel setting so that it doesn’t go out of training distribution. How does SLRL method compares to any continual lifelong RL approaches? A comparison could demonstrate scenarios where focussing on the prior data for exploration clearly beneficial (or detrimental) to the performance of the agent.
- Is a fundamental assumption in SLRL that the reward distribution and state which represent goal should have same distribution as the shown in prior trajectories?  It seems that weighting on Q-values from prior data will induce this assumption and limit the possible scenarios in which the policy will generalize to new settings.

**Ethics Review Area:**

["I don’t know"]

**Limitations:**

The authors have discussed the potential limitations and future scope of SLRL. Some aspects of potential negative societal impact of the  work should be included, like if SLRL is used in home or space-probing robot, what the scenarios will be that it could not handle.

**Strengths And Weaknesses:**

The paper is well-written and structured. The paper discusses the The Q-values from prior dataset allow that generalization to novel unseen scenarios at test time.

However, unlike the motivating example of finding water on Mars where the water would not be found at the same place as a desert on Earth, the experiment setup does not discuss if the goal position would be changed, different from that seen in the prior dataset. The goal remains the same in prior data and “novel” test setting, for example, in pointmass environment at (100, 0), in Franka kitchen with both microwave and cabinet closed, etc.

---

> ### Author Response · Authors · 2022-08-02
> **Response to Reviewer jw3X**
>
> We thank you for your thorough review. We include clarifications and answers to individual questions below. If these responses do not address all of your concerns, please let us know!
>
> > " Unlike the motivating example of finding water on Mars where the water would not be found at the same place as a desert on Earth, the experiment setup does not discuss if the goal position would be changed, different from that seen in the prior dataset."
>
> One of the main contributions of this work is to formalize the SLRL problem, and we aimed to do so in a way that encapsulates many real world situations. In our experimental setup, we do keep the rewards the same between the source and target domains, but this does not mean that the goal state distributions must stay the same. To illustrate an instantiation of SLRL where the goal position is changed, we will add an experimental domain with new goal positions in the final version. We hope that future work can address more severe distribution shifts than our current experimental setup.
>
> > "SLRL is claimed to be a special case of continual or lifelong learning… How does SLRL method compare to any continual lifelong RL approaches?"
>
> Existing continual and lifelong methods are typically not reset-free and are hence not well-suited for SLRL. To illustrate this, we ran a new experiment that evaluates Progressive Nets [1], a continual learning framework that transfers previously learned features, on our Tabletop domain.
>
> |                              | Method               | Avg $\pm$ Std error | Success / 10 | Median |
> |------------------------------|----------------------|---------------------|--------------|--------|
> | Tabletop | Progressive Networks | 159.0k  $\pm$ 21.2k | 3            | 200.0k |
> |                              | QWALE                | 44.4k  $\pm$ 24.6k  | 9            | 8.9k   |
>
> We find that unlike QWALE, progressive nets does not quickly recover from novel situations to complete the desired task. This experiment shows how the SLRL setting may necessitate algorithms different from typical continual RL approaches. We show the results in the Appendix.
>
> > "Is a fundamental assumption in SLRL that the reward distribution and state which represent goal should have same distribution as the shown in prior trajectories?"
>
> In SLRL, the reward between the source and target MDPs is assumed to stay the same, but this does not mean that the goal state distributions must stay the same, since the goal state can be folded into the state space such that changes in the goal state distribution correspond to different initial state distributions. Eventually, we hope that this work may be extended to the case where the reward may be different in the online trial from the prior data. We have added a mention of this in the conclusion in Section 7.
>
> [1] Rusu, Andrei A., et al. "Progressive neural networks." arXiv preprint arXiv:1606.04671 (2016).

---

> ### Author Response · Authors · 2022-08-05
> **Following up**
>
> Hi reviewer jw3X,
>
> Please let us know if our response addresses your concerns. We would be happy to provide further revisions to address any remaining issues. Thank you again!

---

> ### Author Response · Authors · 2022-08-08
> **Checking in**
>
> Hi reviewer jw3X,
>
> We wanted to check in again. Can you let us know if our revisions and response address your concerns? If not, we would be happy to provide further revisions for remaining concerns. Thank you!

---

### Official Review · Reviewer_bBp1 · 2022-07-10

**Rating:** 6
**Confidence:** 4
**Soundness:** 3 good
**Presentation:** 2 fair
**Contribution:** 2 fair

**Summary:**

This paper introduces and tackles the single-life reinforcement learning (SLRL) problem, in which there are no episodic resets available, and the agent has to solve the task in one-shot. The agent has access to prior data which comes from a similar but not the same environment, and this data might not be expert data. The authors propose QWALE, Q-Weighted Adversarial Learning, which extends Adversarial Imitation Learning methods to the general setting in which  the data might not be optimal for the current environment, by weighting the experiences with a pretrained Q-function.

**Questions:**

- Figure 2 positioning is not good. It’s hard to understand what the point of Figure 2 is until you reach Section 7.
- Both PointMass and TableTop training/testing environment sound very similar to the test bed environments used in meta learning papers, where the dynamics or the goals change slightly in the environment. For this reason, I would expect that Meta Learning algorithms like MAML would perform well in environments like tabletop, have the authors tried something similar?
- Figure 5 is never linked, I guess it should be linked somewhere around section 7.3?


**Limitations:**

Overall, this paper introduces SLRL and shows how current algorithms are not capable of autonomously recovering in an environment in which they have to reach the goal but are not provided with resets from bad states. It evaluates current algorithms and proposes a new weighted adversarial learning approach to overcome this problem. QWALE is sound and the idea of weighting the examples based on the Q-function trained in the source environment makes sense.
The main limitation of this work is that it’s very similar to other approaches like the ICLR 2022 paper Autonomous Reinforcement Learning [2] which introduces a similar setting in which environment resets are rare or not available at all, and I am not sure how useful it is to the community.

[2] Sharma, Archit, et al. "Autonomous Reinforcement Learning: Formalism and Benchmarking." ICLR 2022.

**Strengths And Weaknesses:**

Strength
- It introduces and describes single-life reinforcement learning as a specific set of tasks in which an agent has to learn how to recover from mistakes on its own without relying on episodic reset
- Good comparisons to other methods
Weaknesses
- I am not sure how clear it is the importance of SLRL problems. The examples described of a robot exploring another planet or a rescue robot are closely related to the concept of safe reinforcement learning or safe exploration [1], where some constraints have to be respected. Moreover, I would think that similar problems are addressed in classical robotics motion planning literature.

[1] Garcıa, Javier, and Fernando Fernández. "A comprehensive survey on safe reinforcement learning." Journal of Machine Learning Research 16.1 (2015): 1437-1480.

---

> ### Author Response · Authors · 2022-08-02
> **Response to Reviewer bBp1**
>
> We thank you for your detailed feedback. We include clarifications and answers to individual concerns below. In light of these new clarifications, please let us know if you have any remaining concerns. We are happy to answer any further questions you may have.
>
> > "I am not sure how clear it is the importance of SLRL problems. The examples described of a robot exploring another planet or a rescue robot are closely related to the concept of safe reinforcement learning or safe exploration [1], where some constraints have to be respected. Moreover, I would think that similar problems are addressed in classical robotics motion planning literature."
>
> It's true that similar situations to the examples given may be addressed in safe reinforcement learning and in classical motion planning literature. However, safe reinforcement learning typically considers the episodic RL setting with access to resets, and motion planning also makes very different assumptions to our setting, such as giving the agent awareness of all known obstacles during test time. To not overcomplicate our problem setting formalization, we do not specifically account for the case where constraints must be respected, but these may be incorporated into the SLRL setting in future work.
>
> > "The main limitation of this work is that it’s very similar to other approaches like the ICLR 2022 paper Autonomous Reinforcement Learning [2] which introduces a similar setting in which environment resets are rare or not available at all, and I am not sure how useful it is to the community."
>
> We acknowledge that the SLRL setting is similar to the autonomous reinforcement learning in that both do not allow resetting of the environment. However, key to our setting is the goal of completing the task a single time rather than recovering an effective policy that can perform the task repeatedly. In addition, also key to our setting is figuring out how best to leverage prior data, which may differ from the target environment. As a result, our approach, QWALE, is quite different from typical approaches in ARL, which generally do not focus on completing the task as quickly as possible.
>
> To illustrate that ARL methods do not translate well to the SLRL setting, we run MEDAL [1], a recent state-of-the-art ARL method, in the SLRL setting in the Tabletop environment and find that it performs significantly worse than QWALE, as shown in the following table.
>
> |                              | Method | Avg $\pm$ Std error  | Success / 10 | Median |
> |------------------------------|--------|----------------------|--------------|--------|
> | Tabletop | MEDAL  | 176.2k   $\pm$ 15.9k | 3            | 200.0k |
> |                              | QWALE  | 44.4k  $\pm$ 24.6k   | 9            | 8.9k   |
>
>
> We also display these new results in the Appendix.
>
>
> > "Figure 2 positioning is not good. It’s hard to understand what the point of Figure 2 is until you reach Section 7.
>
> We thank you for pointing this out. We have moved this figure and the contents of the section to the Experiments section (now Section 6) in our revision to improve clarity.
>
> > "Both PointMass and TableTop training/testing environment sound very similar to the test bed environments used in meta learning papers, where the dynamics or the goals change slightly in the environment. "
>
> Unlike meta-learning settings, the experimental problems we consider include data from only one source domain rather than from multiple source domains or tasks. Thus, standard meta-RL algorithms are not directly applicable.
>
> > "Figure 5 is never linked, I guess it should be linked somewhere around section 7.3?"
>
> Thank you for pointing this out. We have added a reference to the figure in what is now Section 6.3.
>
> [1] Archit Sharma, Rehaan Ahmad, and Chelsea Finn. "A State-Distribution Matching Approach to Non-Episodic Reinforcement Learning." International Conference on Machine Learning (2022).

---

> ### Author Response · Authors · 2022-08-05
> **Checking in**
>
> Hi reviewer bBp1,
>
> We wanted to follow up to see if the response and revisions address your concerns. Please let us know if you have any further questions. Thank you again!

---

> ### Author Response · Authors · 2022-08-08
> **Following up**
>
> Hi Reviewer bBp1,
>
> We wanted to check in again to see if our revisions and response address your concerns, and if so, whether your evaluation of our paper has changed. If not, we would be happy to provide further revisions. Thank you!

---

### Official Review · Reviewer_muVA · 2022-07-27

**Rating:** 5
**Confidence:** 4
**Soundness:** 2 fair
**Presentation:** 3 good
**Contribution:** 2 fair

**Summary:**

This paper considers the non-episodic RL setting (no environment resets) where an agent is given experience data from a source task and needs to efficiently learn to maximize rewards. They give discussions on why current RL approaches like SAC and GAIL will struggle in this setting (mainly due to distributional shifts from a source task). They then propose QWALE, an "adverserial imitation learning" approach (similar to GAIL) that uses Q-values (pretrained) to weight a discriminator that is learned and used as auxiliary rewards during training. They show experimentally that previous works (including GAIL) indeed struggle in this setting and are outperformed by QWALE.

**Questions:**

- Missing reference in line 248: "Appendix ??"
- $q_\theta$ appears in algorithm 1 and line 243, is critical to the approach, but is never defined. Did you mean $q_\phi$?
- How is $D_{online}$ updated in Alg 1?
- In line 618 of the Appendix you give a different reward function used to speed up training (different from the one in Algorithm 1). What exact reward function was used for training? Was the same reward function used for the GAIL baseline?
- In figure 5 (or the Appendix), please include the trajectories for SAC and GAIL (and compare them with QWALE)
- How sensitive is QWALE to the hyper-parameters involved? A hyper-parameter sweep plot would be useful (since there are no theoretical contributions to strengthen the claims)

**Limitations:**

The authors have adequately addressed most of the limitations (except where mentioned in my review above) and potential negative societal impact of this work.

**Strengths And Weaknesses:**

This is a very interesting paper that proposes a novel algorithm (QWALE) to learn policies in non-episodic discounted settings. I believe this is very relevant to the RL community, and the contributions are significant relative to previous works. Mainly,
- They propose the single life RL setting, which is a novel more realistic non-episodic setting where the agent is given data collected (and policies/value functions learned) from previous experience.
- They provide intuitive explanations for why previous works can struggle in this setting.
- They provide a number of experiments in multiple domains (including complex continuous domains) demonstrating how QWALE out performs previous RL methods like GAIL in the non-episodic setting.

However, I have a couple important concerns on whether the claims made in the paper are well supported. If I understand correctly, the specific main contributions of this work are the new discriminator update rule (eqn 1) and the reward shaping approach (line 7 in algorithm 1). However
- No theory is given about them. For example, how does the weightings added to the discriminator update rule affect the theoretical analysis of GAIL [26]. How does the proposed reward shaping affect optimality?
- No explanation was given for the specific choice of the weightings ($exp(Q(s, a) − b$) and $exp(-(Q(s, a) − b))$. Why is there an exponential in there and what is the effect of b theoretically or experimentally?
- b is claimed to incentivize the agent to move towards states with higher value than its current state. This was not supported theoretically nor experimentally.

---

> ### Author Response · Authors · 2022-08-02
> **Response to Reviewer muVA**
>
> Thank you for your detailed comments. We are glad that you find the paper interesting. Please let us know if the responses below address all of your concerns!
>
> > No theory … For example, how does the weightings added to the discriminator update rule affect the theoretical analysis of GAIL [26]...No explanation was given for the specific choice of the weightings.
>
> We have begun drafting some theoretical justification for QWALE and plan to revise Section 5 with it in the next week. Our initial analysis has led to improvements in the method; in particular, we find that it is better justified to not include the weights on the negative examples when training the discriminator, which has led to an improvement in performance, particularly in the Franka-Kitchen domain. Thank you for helping bring about these improvements. While it's possible that other weighting schemes could also be used, we hope our explanation will provide some justification to the reader for our particular design choice.
>
> > In line 618 of the Appendix you give a different reward function used to speed up training (different from the one in Algorithm 1). What exact reward function was used for training? Was the same reward function used for the GAIL baseline?
>
> We used the auxiliary reward function given in Appendix A.1. for all QWALE and GAIL runs. We have fixed Algorithm 1 to reflect this.
>
> > In figure 5 (or the Appendix), please include the trajectories for SAC and GAIL (and compare them with QWALE)
>
> We have moved the online state visitation plots for SAC from Figure 2 to Figure 5 for a clearer comparison to QWALE. We will include trajectories for GAIL in the Appendix in the final version.
>
> > How sensitive is QWALE to the hyper-parameters involved? A hyper-parameter sweep plot would be useful (since there are no theoretical contributions to strengthen the claims)
>
> Our implementation is built on top of the public implementation of SAC from the EARL benchmark and uses the default hyperparameter values. The only additional hyperparameter used in QWALE is a bias term $b$. In our implementation, we do not need to tune b because we just use the value of the most recent state, but we will provide a hyperparameter sweep of different constant b terms for the Tabletop environment in the Appendix in the final version. Additionally, we will provide more theoretical grounding to our method in our revised Section 5.
>
> > Missing reference in line 248: "Appendix ??"
> > qθ appears in algorithm 1 and line 243, is critical to the approach, but is never defined. Did you mean qϕ?
> > How is Donline updated in Alg 1?
>
> We have made minor typo changes in Section 5 to address these three points. We thank you for your thorough reading of our work and constructive feedback.

---

> ### Author Response · Authors · 2022-08-05
> **Updated submission and checking in**
>
> Hi reviewer muVA,
>
> We have updated the paper with some additional theoretical justification for QWALE. GAIL finds a policy whose occupancy measure minimizes the Jensen-Shannon (JS) divergence to the given prior data. In contrast, particularly because our given prior data may include arbitrarily suboptimal state, we do not want to treat all prior data as equally desirable. Thus, we aim to minimize the JS divergence over the policy's occupancy measure and a target distribution that leads towards task completion. This gives QWALE, where the altered target distribution that the algorithm minimizes the JS divergence over leads to the weighting term added to the discriminator update. We describe this in more detail in Section 5 and Appendix A.1.
>
> We wanted to follow up to see if the response and revisions address your concerns. We would be happy to provide further clarifications and revisions if you have any further questions. Thank you again for all of your detailed comments!

---

> ### Author Response · Authors · 2022-08-08
> **Following up**
>
> Hi Reviewer muVA,
>
> We wanted to check in again. Can you let us know if our revisions and response address your concerns, and if so, whether your evaluation of our paper has changed? If not, we would be happy to provide further revisions. Thank you!

---

### Author Response · Authors · 2022-08-02
**Summary of revisions**

We thank all the reviewers for their thoughtful and thorough comments, and we appreciate the positive assessment of the work from all reviewers. We believe that the feedback received has been very helpful in improving the paper. Based on the comments from all reviewers, we have performed new experiments and revised some of our writing to improve the clarity and structure of the paper.

In summary, our key changes include:
- Additional experiments added to the Appendix, including:
    - 3 additional baseline studies (as suggested by Reviewer bMAN): As expected by Reviewer bMAN, we find that SAC without online learning performs poorly and finetuning SAC with a behavioral cloning loss only helps if the dynamics are not changed.
    - An additional comparison running a reset-free algorithm, MEDAL, as well as an additional comparison with Progressive Neural Networks, a continual learning method, in the SLRL setting in the Tabletop domain. We include these comparisons to demonstrate how the SLRL setting differs from typical autonomous RL and continual learning settings. Both methods perform poorly in SLRL setting, which is unsurprising because SLRL has the goal of completing the desired task a single time as quickly as possible, which is different from the other two settings. Thus, typical reset-free and continual learning algorithms may not be effective in the single-life setting.
- To improve clarity, we have moved the contents of what was formerly Section 5, including the state visitation plots for SAC finetuning, into the Experiments section (now Section 6). Hence, what was Section 6 is now Section 5 (description of QWALE). We removed the plots in what was formerly Figure 4 and moved the results into Table 1 due to space constraints.
- We have added some additional justification for QWALE to Section 5 and Appendix A.1., and our analysis has led to improvements in the method.

We hope that these additional experimental results and revisions have addressed the concerns of the reviewers. If there are any remaining questions, please let us know!

---

### Meta-Review · Area_Chair_48LR · 2022-08-26

**Recommendation:** Accept
**Confidence:** Less certain

**Metareview:**

The paper introduces a new formulation for single life reinforcement learning which is interesting. Moreover, an algorithm is presented for solving this RL scenario. The paper was evaluated positively by all reviewers. The 2 borderline reviews main concerns were:
- missing theoretical evidence / motivation for the algorithm (Riviewer muVA): This concern has been mostly addressed by the authors. They motivate their choice of the weights, but how to incorporate the weights into the algorithm is clear to me on an intuition level, but not so much backed on theory.
- the algorithm was not illustrated for scenarios with changing goals (Reviewer jw3X): This concern was addressed by the rebuttal.

Unfortunately, the two reviewers with the borderline scores  did not respond to the rebuttal, but I think their concerns have been mostly addressed and they should have raised their score. Hence, I recommend accepting the paper.

**Award:**

No

---

### Decision · Program_Chairs · 2022-09-14

Accept